# Pan-genome insights into type VI secretion systems and their functional repertoires in *Enterobacter*

Zhihan Peng,[1] Ning Zhu,[1] Wenjing Yi,[1] Lili Jiang,[2] Tingting Dong,[1] Ruihong Wu,[3] Shanshan Jia,[1] Xiaohan Guo,[3] Zhaoqing Luo,[4] Qingtian Guan[1]

**ABSTRACT** The *Enterobacter* genus contains 23 species that include common nosocomial pathogens capable of causing a wide variety of infections. We obtained all available *Enterobacter* genomes and retained 4,805 high-quality genomes after quality control. Genome sequencing analysis of *Enterobacter* species revealed the presence of type VI secretion systems (T6SS) in these bacteria, but systematic analysis and comparison of these systems among different species are limited. We found that these bacteria code for three distinct types of T6SS, each with a unique set of diverse predicted effectors. Whereas at least 14 effectors are found in each strain, the number of immunity proteins is considerably fewer. By demonstrating a correlation between the abundance of known T6SS-associated proteins and the presence of T6SS, we proposed a comparative genomics model to evaluate the correlation between unknown T6SS-associated ortholog proteins and T6SSs. Among the homologous groups most strongly associated with T6SS, we potentially identified several effectors. It is conceivable that our methodology could be scaled to survey additional bacterial genera for novel T6SS effectors, thereby providing fresh perspectives and directions for subsequent biological experiments.

**IMPORTANCE** *Enterobacter* species are important human pathogens that can cause severe conditions like pneumonia, urinary tract infections, and bacteremia. As Gram-negative bacteria, they frequently carry diverse T6SS loci, which are often associated with bacterial virulence and are also one of the important causes of bacterial infection. T6SS effectors play a critical role in interbacterial competition and virulence during infection. VgrG proteins are essential T6SS components that form the spike structure and mediate effector delivery, making them critical for bacterial competition and virulence. However, systematic studies on their distribution and function remain limited. Here, we analyzed all available high-quality *Enterobacter* genomes and revealed that T6SS diversity is shaped by both species' evolution and horizontal gene transfer (HGT). We proved that it is feasible to measure the biological relevance of unknown functional proteins to the T6SS through statistical analyses. This high-throughput approach provides a new perspective for future research on T6SS functionality, especially in *Enterobacter*.

**KEYWORDS** *Enterobacter*, type VI secretion system, effectors, bacterial competition, evolutionary adaptation

T he T6SS is a widespread multiprotein machinery that plays important roles in interbacterial interactions, symbiosis, virulence, and stress resistance (1, 2). Based on TssB phylogeny, T6SS is classified into four clades (i–iv) (3, 4). T6SS comprises 13 (TssA-TssM) core components (5) organized into the membrane complex (TssJ, L, M), the baseplate (TssE, F, G, K), the contractile sheath (TssB, C), and the distal end (TssA) (6). TssI, also known as VgrG (valine-glycine repeat G) proteins, are essential T6SS components

**Peer Reviewer** Min Yue, Hangzhou Institute for Advanced Study, Hangzhou, Zhejiang, China

Address correspondence to Qingtian Guan, Qingtian_guan@jlu.edu.cn, or Zhaoqing Luo, luoz@jlu.edu.cn.

The authors declare no conflict of interest.

See the funding table on p. 15.

that form the spike structure and mediate effector delivery (7). T6SS effectors are often delivered by covalent or non-covalent association with VgrG, which can carry C-terminal extensions that function as effectors themselves or serve as adaptors to recruit additional toxic proteins for translocation (8). The contraction of the sheath forces the inner tube, with its associated effector proteins or nucleic acid, into the target cells (9) (Fig. 1A).

In fact, the first identified T6SS effector, VgrG-1, was discovered and encoded alongside *vgrG* within an open reading frame (2). However, some non-toxic C-terminal extensions of VgrG, such as DUF2345, are not predicted to exert toxic activity and may also serve as "loading platforms" for effectors (10). Effectors have also been found that can function as components of the secretion apparatus and are, therefore, often encoded downstream of *vgrG,* that is, in the *vgrG*-proximal region (11). The *vgrG*-proximal region is defined as the three genes immediately downstream of the *vgrG* in the same contig in the direction of *vgrG* transcription; if fewer than three genes are present downstream on the contig, all downstream genes within the same contig are considered. For these reasons, *vgrG* genes serve as an essential resource for effector identification and were demonstrated in several bacteria, such as *Escherichia coli* (12), *Agrobacterium tumefaciens* (13), *Pseudomonas aeruginosa* (14), and *Vibrio parahaemolyticus* (15).

T6SS effectors employ diverse mechanisms—such as lipase and nuclease activities or inhibition of cell‐wall and protein synthesis—to kill competing cells. For example, the *Yersinia pseudotuberculosis* T6SS-3 kills competitive cells by secreting the nuclease effector Tce1 into the extracellular medium, entering target cells, which requires the outer membrane proteins OmpF and BtuB (16). Tle1 from *Aeromonas hydrophila* strain NJ-35 is a membrane-disrupting effector that degrades the lipid bilayer of the target cell (17). *Enterobacter bugandensis* (18) and Enterohemorrhagic *Escherichia coli* secrete the effector KatN into host cells when the bacteria sense oxidative stress in the phagosome (19). Beyond interbacterial killing, T6SSs also mediate metal ion uptake (20), oxidative stress defense (21), and biofilm formation (22). A T6SS immunity protein (or "cognate immunity protein" in the context of the type VI secretion system) is a bacterial protein that neutralizes or inhibits the toxic activity of a specific T6SS effector, thereby protecting the producing (or "self") cell from self-intoxication or sibling-strain killing (23).

*Enterobacter* spp. are notable nosocomial pathogens associated with multidrug-resistant infections and also inhabit diverse environments from soil to the human gut (24). The most commonly isolated species are *Enterobacter hormaechei*, followed by *Enterobacter cloacae*. However, not all *Enterobacter* species are known to infect humans—for example, *Enterobacter soli* (25) and *Enterobacter lignolyticus* (26) are environmental strains, and to date, there is no evidence that they are pathogenic. At least four types of secretion systems, including T1SS, T2SS, T4SS, and T6SS, have been identified in *Enterobacter cloacae* (27–29). The diverse niches and competitive pressure experienced by the *Enterobacter* genus bring variations in the genomic arrangement of its T6SSs. Two distinct systems have been functionally validated in *E. cloacae* strain 13047. One functions as an antibacterial weapon, and the second plays an essential role in biofilm formation and adherence to eukaryotic cells (30). The arsenal of T6SS effectors, which has a dynamic composition among strains, is understudied. Currently, only four T6SS effectors (RhsA, RhsB, Tae4, and Tle) have been reported in *Enterobacter*, all of which were identified in *E. cloacae* 13047 (18). The RhsA and RhsB effectors target both prokaryotic and eukaryotic cells (31). The Tae4 confers a slight growth advantage in competition (32).

Effector diversity in T6SS provides bacteria with a broader arsenal of toxic activities, enabling them to outcompete diverse microbial rivals, adapt to different ecological niches, and enhance survival in polymicrobial environments (33). To gain a competitive advantage, a bacterium enhances the diversity of its effectors through horizontal gene transfer (HGT), convergent evolution, or gene duplication (34). Existing computational approaches on whole-genome data either focus on genes associated with T6SS loci, such as *vgrG*-proximal genes (35), those located within T6SS clusters and auxiliary modules

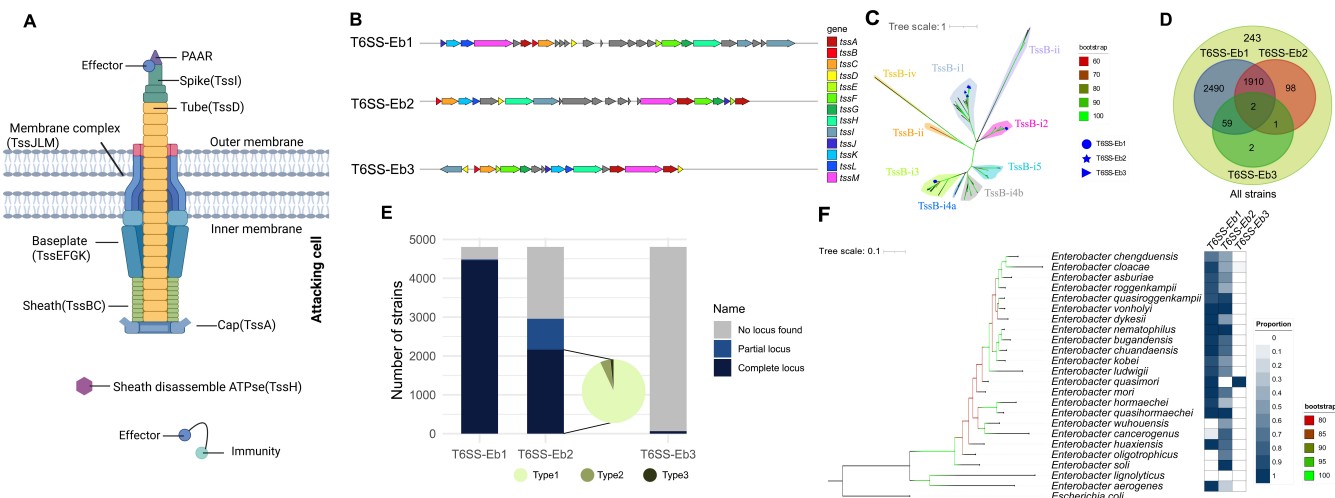

**FIG 1** Gene organization and phylogenetic relationships of the three T6SSs distributed across *Enterobacter*. (A) Schematic representation of the T6SS. The different core components are labeled. (B) A schematic representation of three different T6SS gene cluster structures. All three types of T6SS contain 13 core components (TssA-TssM), indicated in different colors. Gray regions represent the other genes. The arrow length is proportional to the gene length. (C) Phylogenetic tree was constructed using 117 amino acid sequences of TssB in *Enterobacter* strains and SecReT6. Twenty-seven TssB sequences from our three loci (Eb1/Eb2/Eb3) are highlighted by blue circle, star, and triangle; the remaining 90 sequences (from SecReT6) are classified into clades i–iv. Background colors represent different types, and textual annotations are provided for clarity. The scale bar represents the number of amino acid substitutions per site. (D) Venn diagram representing the distribution of T6SS types in each strain. (E) The histogram shows the distribution of the three T6SS types across all *Enterobacter* species. From left to right, the distribution of T6SS-Eb1, T6SS-Eb2, and T6SS-Eb3 is displayed. Dark blue indicates the presence of the full T6SS structure in 4,805 strains of *Enterobacter*. A partial T6SS, defined as lacking more than one core component but retaining at least five, with the same arrangement of core components as in the complete cluster, is shown in light blue. Subtypes of T6SS, containing all 13 core components but missing or duplicating one compared to the most prevalent T6SS, are shown in a pie chart. Types and subtypes with fewer than 50 instances are not counted. (F) The phylogenetic tree was generated using a representative genome selected from each species, with the standard strain of *Escherichia coli* as an outgroup. The branch colors indicate Bootstrap values from 80 to 100. The heatmap on the right represents the distribution of the three T6SS types across species, where each column represents a T6SS and each row a species. From dark to light, the color intensity reflects the proportion of strains in each species with a complete T6SS. The scale bar represents the number of amino acid substitutions per site.

(36), or rely on strict comparative genomics on strict comparative genomics to identify genes uniquely present in T6SS-containing genomes (37). While these strategies have been widely applied to map T6SS loci and predict candidate effectors (18), they often fail in the absence of prior knowledge or in genomes harboring multiple T6SS loci. Therefore, novel methods are needed to overcome these limitations.

To address this gap, we performed a computational analysis of T6SSs in *Enterobacter* spp. Our results revealed three different loci and a novel widespread T6SS associated with species across the genus. For the genomic region most enriched in T6SS effectors, namely the proteins encoded downstream of *vgrG*, we investigated their distribution patterns. Our analysis revealed a wide variety of conserved protein domains in this region, while 49.1% of the protein sequences lacked any recognizable domains. These analyses have extended our appreciation of T6SS effectors of the genus. Finally, we proved that there are significant differences between genomes with or without T6SS in the number of T6SS-associated proteins. We then performed a statistical approach to evaluate the relationship between the protein groups and the T6SS based on this difference, uncovering novel T6SS-associated ortholog groups. This analysis revealed several T6SS-associated loci or orphan genes that likely code for novel effectors.

## RESULTS

### Tripartite T6SS system architecture with phylogenetic divergence

To obtain high-quality genomes, we performed stringent quality control on all available genomes, resulting in a final data set of 4,805 genomes after filtering. These

genomes were subsequently subjected to taxonomic assignment, confirming consistent species-level classifications (Table S1). To conduct a comprehensive analysis of T6SS loci in *Enterobacter*, we identified three distinct major T6SS loci (T6SS-Eb1, T6SS-Eb2, and T6SS-Eb3) across all 4,805 high-quality genomes, each exhibiting distinct structural configurations (Fig. 1B). Notably, T6SS loci are not present in all strains: complete T6SS-Eb1 is found in 4,488 strains, complete T6SS-Eb2 in 2,011 strains, and complete T6SS-Eb3 in 64 strains (Fig. 1D). Subsequently, we compared the protein sequence similarity among the same components of three different T6SS types. Since the classification of T6SS types in our study was based on the arrangement of core components, we also examined the sequence similarity of homologous components within the same T6SS type. Multiple sequence alignment results revealed that the same T6SS variability was minimal, whereas their homologs across different T6SS types exhibited low or undetectable sequence similarity (Fig. S1). That means, except for gene order rearrangement, syntenic differences also exist in disparate T6SS. Notably, we identified two subtypes for T6SS-Eb2, further emphasizing its diversity (Fig. S2). Pairwise correlation analysis revealed no significant correlation between T6SS-Eb1, T6SS-Eb2, and T6SS-Eb3, suggesting that these loci are not strongly associated with each other (Table S2). Phylogenetic reconstruction of the TssB sequences further classified these loci into established T6SS clades: T6SS-Eb1, T6SS-Eb2, and T6SS-Eb3 belong to T6SS-i3, T6SS-i2, and T6SS-i1, respectively (Fig. 1C).

T6SS-Eb3, a previously unreported T6SS locus, shares significant sequence similarities (34%–86% amino acid identity) with the T6SS locus of *E. coli* K1 (Fig. S3), which has been implicated in anti-eukaryotic activity, effector translocation, and adherence functions (38). The results of the BLAST (Basic Local Alignment Search Tool) search exclude the possibility of contamination by *E. coli* genomes (Table S3). To further confirm that the high similarity between these two loci is the result of HGT between *E. coli* and *Enterobacter*, we first constructed a single-gene phylogenetic tree based on all TssB sequences from the *Enterobacteriaceae* family (Fig. S4A). To increase phylogenetic depth, we further conducted a multi-locus phylogenetic analysis using the core T6SS components shared across *Enterobacteriaceae* T6SS loci (Fig. S4B). Systematic evolutionary analyses of these phylogenies support this conclusion.

The distribution of T6SS loci across the *Enterobacter* genus revealed notable patterns. T6SS-Eb1 is the most prevalent locus in *Enterobacter* spp., present in more than 92.84% of sequenced strains (4,461/4,805) across 19 species (Fig. 1D). T6SS-Eb2 was found in 2,011 strains across 21 species within the genus, distributed across a larger number of species, but is less prevalent as the number of strains that possess the T6SS-Eb2 is less than that of T6SS-Eb1. In contrast, T6SS-Eb3 was found in 1.3% of total assemblies (64/4,805). Regarding locus completeness, T6SS-Eb2 exhibits lower conservation compared to T6SS-Eb1 and T6SS-Eb3, with 39.5% of T6SS-Eb2 cases being incomplete, versus only 0.07% of T6SS-Eb1 cases. The presence of additional subtypes further underscores this disparity (Fig. 1E).

Phylogenetic analysis revealed that basal species of the genus, such as *Enterobacter wuhouensis*, *Enterobacter cancerogenus*, *Enterobacter oligotrophicus*, *E. soli*, and *E. lignolyticus* (Fig. 1F), exhibited a lower prevalence of T6SS-Eb1. In comparison, T6SS-Eb2 was universally present across the genus, except for *E. lignolyticus* and *Enterobacter quasimori*. T6SS-Eb3 showed a limited distribution and was found in strains of *E. bugandensis* (2/163), *E. cloacae* (29/534), and *Enterobacter hormaechei* (31/3,201), and was present in 100% of *E. quasimori* strains (2/2). This pattern highlights the complex evolutionary dynamics of T6SS in *Enterobacter*, suggesting distinct selective pressures and functional specializations shaping its distribution across different species.

## Diversity and distribution of T6SS effectors, regulators, and immunity proteins in *Enterobacter*

To characterize T6SS effectors, regulators, and immunity proteins in *Enterobacter*, we searched for homologous proteins across genomes to identify all T6SS-associated

proteins, including orphans. This approach identified a total of 115,718 effectors homologous. Notably, only 7,412 proteins (6.4%) were found within the three genes downstream of *vgrG*, indicating a significant portion of the effector repertoire remains unidentified using traditional approaches that focus on *vgrG*-associated loci. Similarly, only a small fraction (3.9%) of proteins homologous to known immunity proteins are located downstream of *vgrG*. To verify that our sampling effort was sufficient, we analyzed the rarefaction curve depicting the change in effector diversity as the number of strains increased. The curve reached a plateau, indicating that the sampling effort was adequate to capture most effectors and immunity proteins of *Enterobacter* (Fig. S5).

To explore the relationship of VgrGs and T6SS, we group the genomes based on the presence or absence of different loci and quantify the effectors within each group. We found that the number of *vgrG* genes differs significantly among the groups. Further analysis revealed that this significant difference only exists between strains with and without T6SS loci, indicating that *vgrG* is considerably more prevalent in genomes containing T6SS loci, regardless of whether they possess one or two loci. (Fig. 2A). To systematically identify T6SS effector genes encoded downstream of *vgrG*, we analyzed 14,812 *vgrG*-proximal regions, encompassing a total of 41,663 proteins, and identified conserved protein domains. Among these, 5,489 proteins contained the PAAR domain, making it the most prevalent domain downstream of *vgrG*. Additionally, 4,191 proteins possessed the DcrB domain, essential for forming complexes between *vgrG* and their specific N-terminal PAAR-domain-containing effectors (39). Finally, 2,762 proteins were found to have the RHS domain (Fig. 2B).

We further grouped the effectors based on homologous proteins to further analyze their distribution and characteristics. A total of 76 effector homologs have been identified (Fig. 3A), including six core effectors that are universally present in different species, and seven effectors are unique to a particular species (Fig. 3B). Among these effectors, EFF01477 is the most prevalent effector that is present across the genus. The effector's translocation, which is T6SS-dependent, was first identified by the macrophage infection-responsive *Edwardsiella piscicida* genes (40). In the *Enterobacter* genus, an average of 31 different effectors is found per species, with a median of 24. As the second largest sequencing data available species, *E. cloacae* harbors the highest diversity of effectors, with 63 distinct protein families identified in its genomes, including two unique effectors absent in other *Enterobacter* species. One is the T6SS effector with DNase

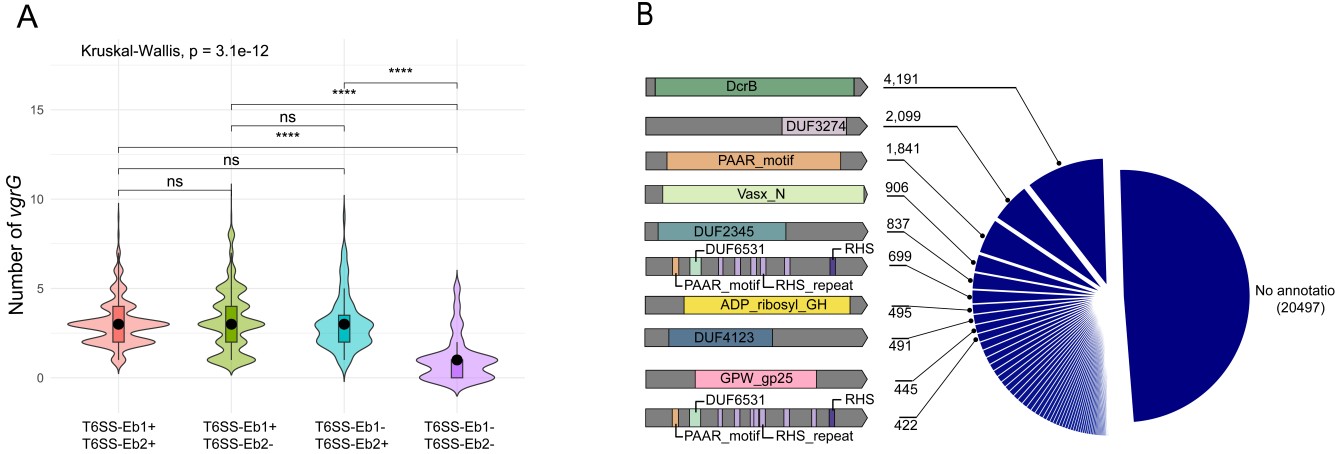

**FIG 2** The *vgrG* contents in *Enterobacter* and its downstream proteins. (A) The violin plot illustrates the distribution of *vgrG* contents across four genome groups from left to right: the first group is genomes with both T6SS-Eb1 and T6SS-Eb2, the second group is genomes with only T6SS-Eb1, the third group is genomes with only T6SS-Eb2, and the fourth group is genomes without a complete T6SS. The Kruskal-Wallis test results indicate that genomes without T6SS have significantly lower *vgrG* content, while genomes with one or two complete T6SS show no significant difference in *vgrG* content. (B) Systematic analysis of genetic systems encoded downstream of *vgrG* in all *Enterobacter* genomes. Pie chart of the 10 most frequent Pfam domains found within three genes downstream of *vgrG* (not to scale).

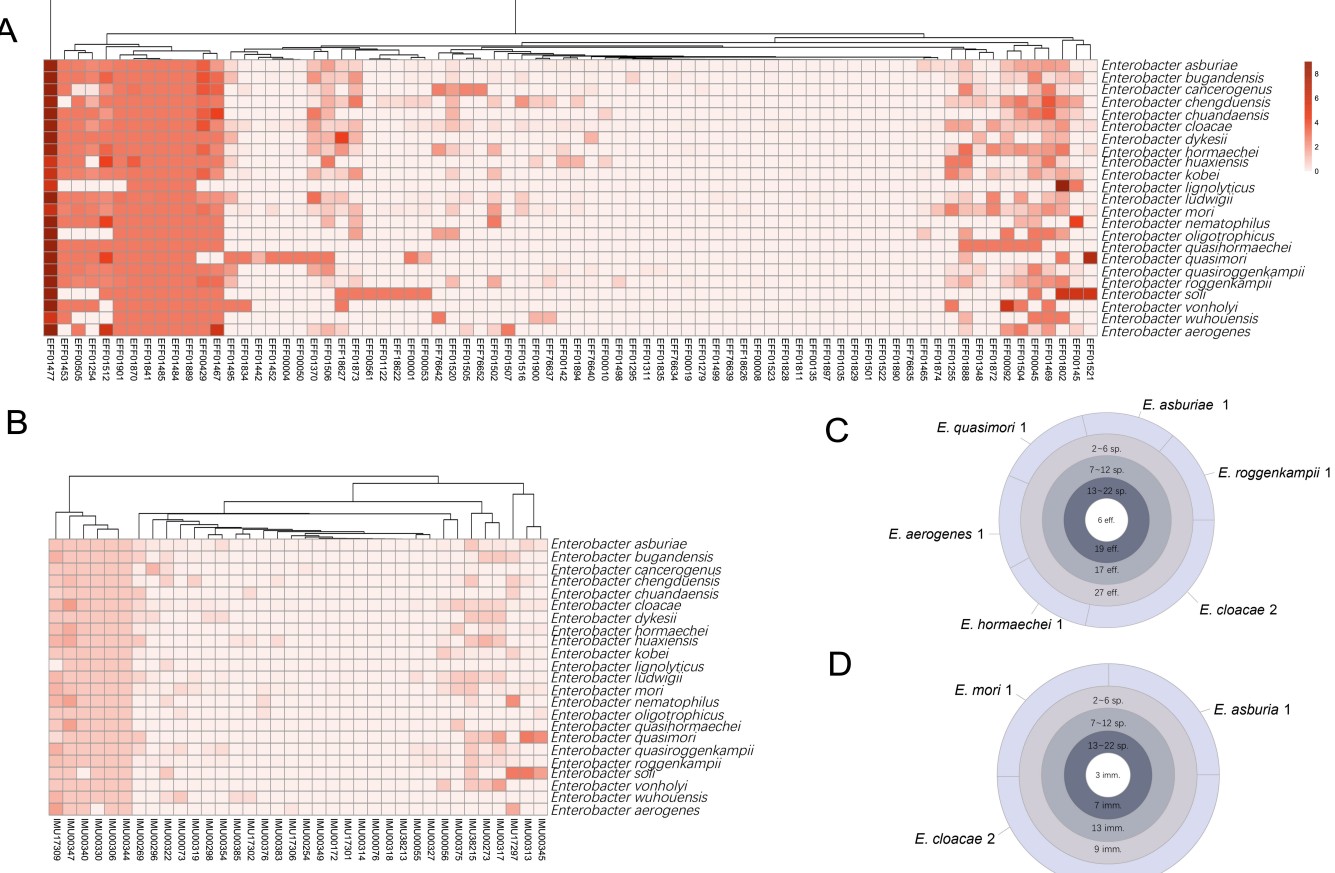

**FIG 3** Distribution and diversity of T6SE and T6SI in *Enterobacter*. (A and C) The heatmaps display the abundance and category of effectors and immunity proteins across species, with color intensity representing abundance. Scale indicates the number of homologous genes. (B) The innermost circle represents 6 effectors found in all species; the second layer shows 19 effectors present in at least 13 but no more than 22 species; and the third and fourth layers represent effectors found in fewer species. The outermost layer shows species-specific immunity proteins, with two found only in *E. cloacae*, one in *E. hormaechei*, one in *E. aerogenes*, one in *E. quasimori*, one in *Enterobacter roggenkampii*, and one in *Enterobacter asburiae*. (D) The innermost circle represents three immunity proteins found in all species, the second layer shows eight immunity proteins present in at least 15 but no more than 22 species, and the third and fourth layers represent proteins found in fewer species. The outermost layer shows species-specific immunity proteins, with three found only in *E. cloacae*, one in *Enterobacter mori*, and one in *E. asburiae*.

activity (named Tde), identified in *A. tumefaciens* strain 1D1609 (41); the other is the type VI lipase effector (Tle), identified in *Serratia marcescens* subsp. marcescens Db11 (42). Surprisingly, eight effector homologs were identified in the genome of *E. lignolyticus*, which lacks any complete T6SS cluster, suggesting the presence of alternative secretion systems yet undiscovered, or that these effector proteins could have functions unrelated to T6SS. These widely distributed and diverse effectors confer stronger bacterial competition and virulence to *Enterobacter* (43).

In contrast to the 76 effector protein families identified in *Enterobacter*, only 36 types of immunity proteins have been detected, reflecting the potential cross-neutralization capabilities of the immune proteins or some non-immunity-mediated protection mechanisms. They are categorized into two groups: one widely distributed across all species and another distinctly present in specific species (Fig. 3C), which reflects the cross-neutralization of the immune proteins. More concretely, three of them exist in all species; they are IMU00347, IMU00344, and IMU00306. IMU00347, encoded by *Bradyrhizobium japonicum* strain J5, is associated with the Type i3 T6SS (44), categorized as T6SS-Eb1 in our study. This protein likely neutralizes the toxic activity of its cognate effector EFF01811, ensuring protection against self-intoxication and potential attacks

from neighboring bacteria. Interestingly, although IMU00306 and IMU00344 share no homology and are not adjacent to each other within each genome, they exhibit identical distribution patterns across *Enterobacter* species, suggesting that they may function synergistically or collectively contribute to immunity. *E. cloacae* harbor the most diverse immunity proteins, with a total of 32, two of which are unique to the species (IMU00318 and IMU38213). Regarding the species-specific immunity proteins, among these, four immunity proteins are uniquely present in one species (Fig. 3D), indicating that these proteins may provide niche-specific protection.

A highly diverse set of regulators has been identified in *Enterobacter*, with a total of 174 distinct types. Among these, 103 are present in all species, while only 13 are unique to a single species. The regulators found in all species can be categorized into two distinct groups: one group, like REG0321, is broadly distributed and abundant across all species; REG0321 was first found in *Pectobacterium brasiliense*, and its absence leads to a significant decrease in the transcription of 23 T6SS-related genes (45). The other group, exemplified by REG0370, is consistently present but at lower levels across species, and it enhances the T6SS-mediated killing of *Acinetobacter baumannii* (46) (Fig. S6). However, regulators are a broadly functioning class of proteins that participate in various cellular activities beyond T6SS-related functions, which partly explains the large number of identified regulators (47). When we increase the cutoff for homologous protein identification, the number of detected regulators decreases significantly. For example, when the cutoff is set to 80% identity and 80% coverage, only 42 different regulators can be identified, and REG0370 is no longer detected in *Enterobacter*.

## T6SS-associated ortholog groups and related functions

We hypothesize that the number of T6SS-associated genes is more abundant in genomes with the presence of T6SS than in genomes without it, leading us to refer to these T6SS-associated ortholog groups as TAOGs. To test the hypothesis, we explore the distribution of known T6SS-associated protein families, identified through homology searches, in the genome with or without T6SS-Eb1 (T6SS-Eb1+ vs T6SS-Eb1−). We selected *E. hormaechei* and *E. cloacae* as the target species because they are the most prevalent human pathogens within the genus and possess the largest genomes in *Enterobacter*.

The normalized number of effectors shows a significant difference in distribution between the two groups, as determined (Fig. 4A). The distribution of effectors is noticeably skewed toward T6SS-Eb1+ strains (Fig. 4B), indicating that T6SS-Eb1+ consistently harbors a higher number of effectors than T6SS-Eb1− strains. This pattern suggests a strong association between the presence of T6SS-Eb1 and an increased repertoire of effectors, potentially reflecting its role in interbacterial competition or host interactions, which is T6SS-dependent.

To further analyze the relationship between the abundance of T6SS-associated proteins and the presence of a complete T6SS in bacterial strains, we conducted a similar analysis for immunity proteins and regulators. All results indicated that strains possessing a T6SS-Eb1 exhibited significantly higher levels of associated proteins than those lacking T6SS-Eb1, including not only T6SS effectors, but also immunity proteins and regulators in both *E. hormaechei* and *E. cloacae*. Likewise, we examined whether these three types of proteins are correlated with the presence of T6SS-Eb2 in the strains. In contrast, this trend was less pronounced for T6SS-Eb2 (Fig. S7). For example, while the number of effectors was higher in the genomes with T6SS-Eb2+ compared to T6SS-Eb2− in *E. hormaechei*, the abundance of immunity proteins did not exhibit the same pattern, with immunity proteins being more abundant in T6SS-Eb2− strains. A similar pattern was observed in *E. cloacae*. This discrepancy is likely due to the influence of T6SS-Eb1, as genomes lacking T6SS-Eb2 are more likely to contain a T6SS-Eb1 locus (2549/1912). Therefore, we excluded T6SS-Eb2 from the downstream analysis. This association study demonstrates that TAOGs are not simply absent or present in strains with or without T6SS but rather exhibit significant quantitative differences between the two groups.

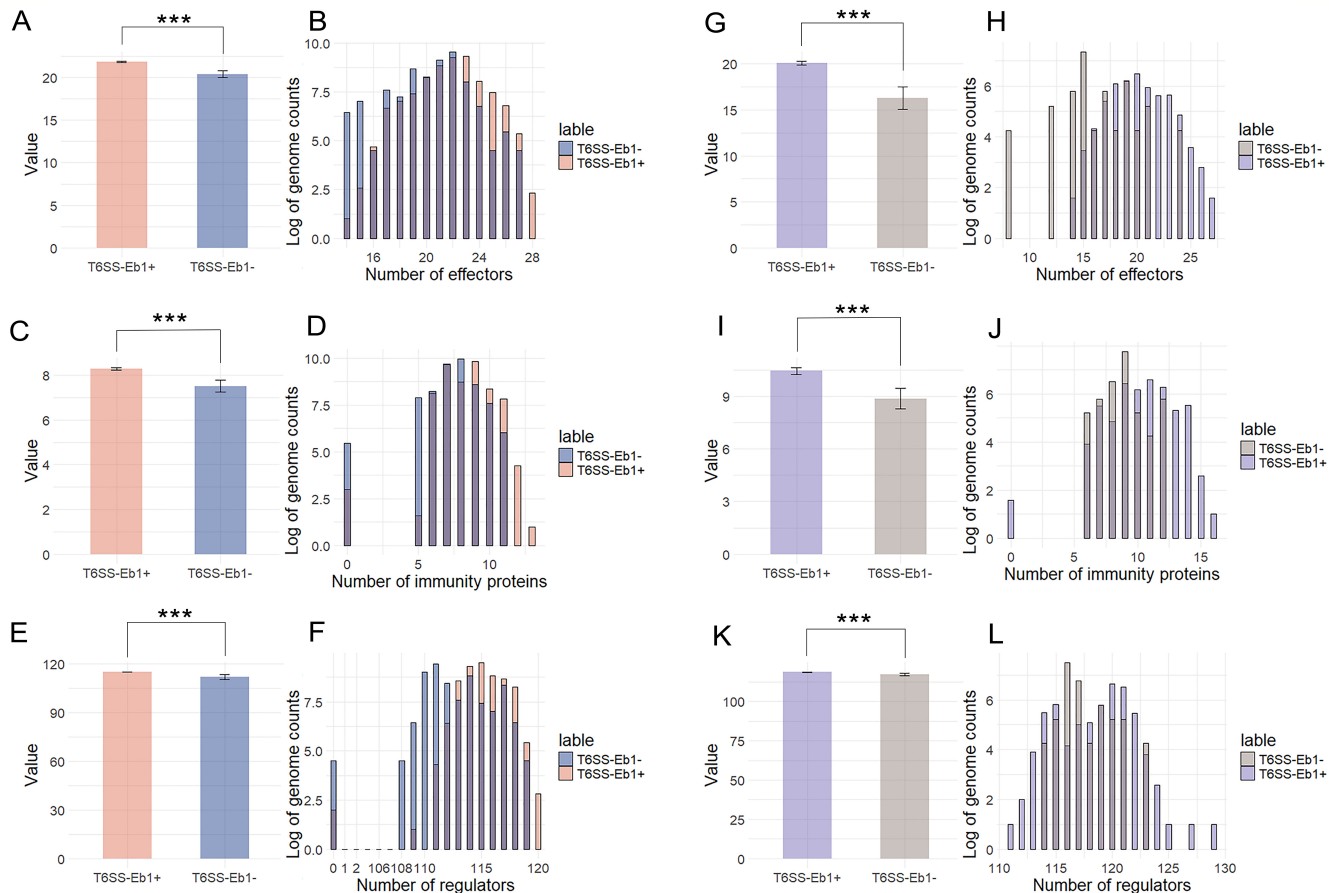

**FIG 4** T6SS-associated proteins are significantly higher in strains with intact T6SS-Eb1 than those without T6SS-Eb1. (A, C, and E) The histogram plots given the differences in effector (A), immunity protein (C), and regulator (E) counts in *E. hormaechei* were used as a representative to analyze whether the presence of T6SS-Eb1 correlates with the abundance of related proteins. The *x*-axis shows the average number of these proteins in genomes with (red) and without (blue) T6SS-Eb1. A statistically significant difference was observed between the two groups (***P value < 0.001), as determined by a nonpaired Student's *t*-test. (B, D, and F) The bar graphs represent the distribution in *E. hormaechei* of effector (B), immunity protein (D), and regulator (F), with the *x*-axis showing the number of effectors and the *y*-axis showing the logarithmic number of genomes. (G, I, and K) The histogram plots give the differences in effector (G), immunity protein (I), and regulator (K) in *E. cloacae,* which were used as a representative to analyze whether the presence of T6SS-Eb1 correlates with the abundance of related proteins. The *x*-axis shows the average number of these proteins in genomes with (purple) and without (gray) T6SS-Eb1. A statistically significant difference was observed between the two groups (***P value < 0.001), as determined by a nonpaired Student's *t*-test. (H, J, and L) The bar graphs represent the distribution in *E. cloacae* of effector (H), immunity protein (J), and regulator (L), with the *x*-axis showing the number of effectors and the y-axis showing the logarithmic number of genomes.

## Potential effectors and unknown functional gene clusters in TAOG

To establish an active genomics pipeline for detecting potential TAOGs, all of the proteins of *Enterobacter* strains have been clustered as putative homologous groups. For T6SS-Eb1, we divided the genome data set into T6SS-Eb1+ (4,488 genomes) and T6SS-Eb1− (293 genomes) classes. Only genomes that contain complete or no T6SS-Eb1 are taken into consideration. Groups were evaluated on how relevant they are to T6SS-Eb1 via a presence/absence ratio (P/A), and significance was confirmed through the chi-square test. With this $log_2$ P/A value of more than 2.4 and this *P* value of the chi-square test less than 0.05, 600 groups containing almost all T6SS core components were selected.

Not surprisingly, many known T6SS effectors and immunity proteins (P/A, chi-square, *P* value) are among the TAOGs (Fig. 5A). The P/A values of known T6SS effectors and immunity proteins range from 2.414 to 3.642, with chi-square *P* values all below 0.034. The highest P/A value is found in Cluster_58461, which has been identified as

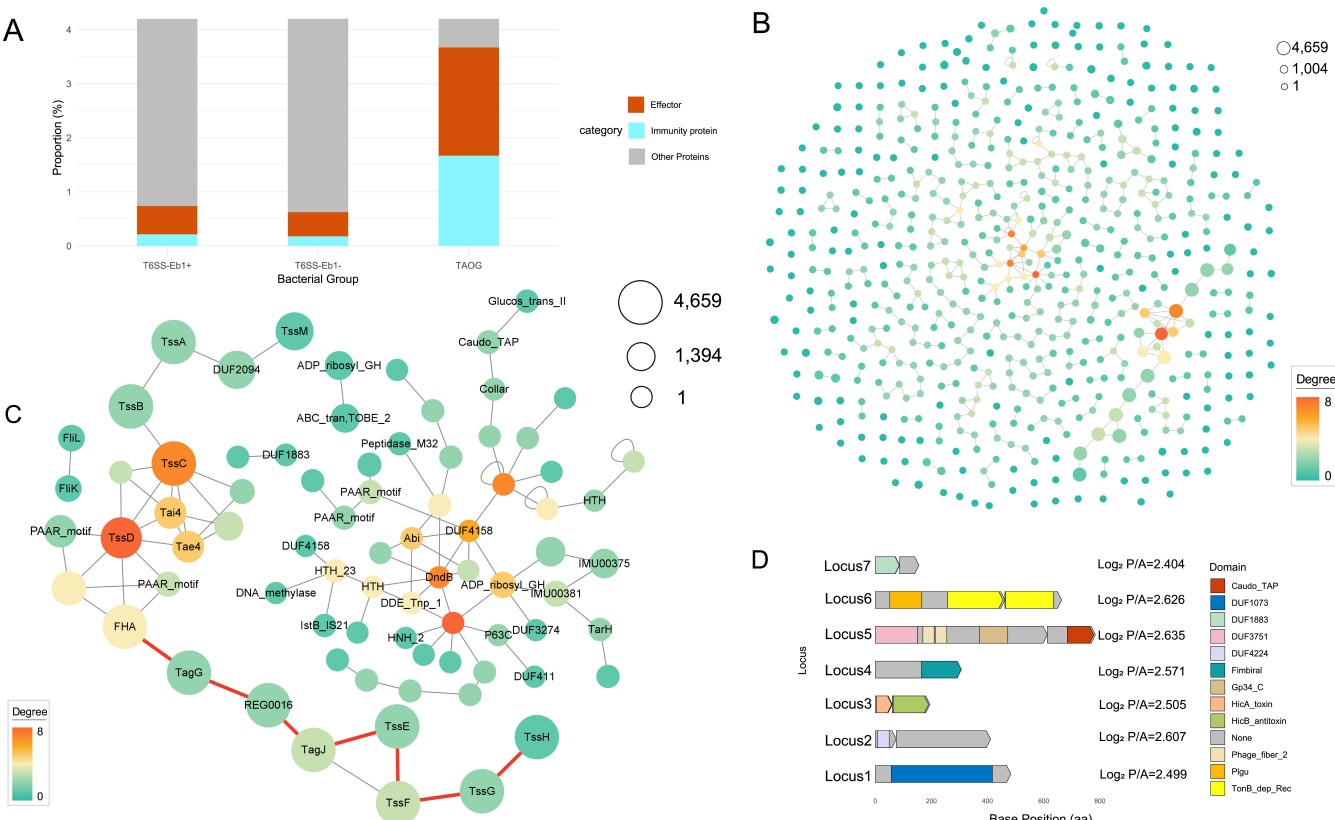

**FIG 5** The function and distribution of T6SS-associated ortholog groups in *Enterobacter*. (A) The histogram shows the proportion of effectors and immunity proteins in T6SS-Eb1+, T6SS-Eb1−, and TAOG. (B) The homologous protein groups with high correlations associated with T6SS-Eb1 were visualized as circles, if proteins from different clusters are encoded adjacent in the genome. Nodes with more connections are larger and darker, reflecting greater cluster size and connectivity. Orphan clusters have no adjacencies to other clusters in the genome. (C) The red line highlights the most frequently connected locus. (D) Some putative effectors and effector-immunity (E-I) pairs in TAOG. Proteins are shown to scale as gene cassettes, colored by their predicted domains.

Tae4. Within the TAOGs associated with T6SS-Eb1, 12 core components are represented alongside the aforementioned experimentally supported T6SS-related proteins. This distribution suggests that some uncharacterized proteins with potential relevance to T6SS may also be present within these groups. However, this possibility arises solely from statistical association. Accordingly, further analyses, such as the identification of conserved domains or other mechanistic features, are required to substantiate any functional relationship between these TAOGs and T6SS.

A total of 122,192 homologous protein families were identified, of which 600 protein family groups were selected as the most relevant groups with T6SS-Eb1. Four hundred twenty-one clusters are linked to each other, and 179 clusters are orphans (Fig. 5B). To infer the potential functions of TAOGs computationally, we analyzed the connectivity of clusters within the genome. We further examined the TAOGs that are associated with specific genomes, which might be a locus that functionally relates to T6SS; among the clusters associated with T6SS-Eb1, 421 out of 600 were found to be connected to other homologous families (Fig. 5C). The most frequently connected locus spans from Cluster_2196 to Cluster_6428 (indicated by a red line), which belongs to T6SS-Eb1, corresponding specifically to the *tssH*, *tssG*, *tssF*, *tssE*, *tagG*, and FHA region. The Cluster_64950, as the highest degree node in connection, is also notable, which means it links with the most different clusters, reciprocal to *tssD*. Cluster_64950, identified as *tssD*, is the highest-degree node, linking to diverse clusters. Specifically, *tssD* forms a locus with Tae4 (Cluster_64148), a T6SS effector, and its cognate immunity protein Tai4 (Cluster_58461). While this Tae4-Tai4-TssD locus is widespread in genomes, it is

not universal: *tssD* can also associate with alternative effector-immunity (E-I) pairs or other T6SS-related components in some instances, such as the loci TssD-Cluster_72226 or TssD-Cluster_103957, which are annotated as hypothetical proteins and are likely to function as T6SEs or auxiliary T6SS components. In another connection network, some noteworthy clusters also emerge. For example, Cluster_85253, which contains a DndB domain involved in the regulation of DNA modifications (48), is linked to multiple Helix-turn-helix domains, suggesting that T6SS-associated proteins may also participate in DNA modification. Additionally, in this network, proteins belonging to Cluster_101935 all contain an abortive infection domain and are consistently linked to Cluster_4389, indicating that this may be an unrecognized E-I pair. Some paired clusters, such as Cluster_16610 and Cluster_13988, are also not to be sniffed. Cluster_13988 contains an ABC transporter domain and a TOBE_2 domain, suggesting a role in substrate transport and regulation. Cluster_16610 features an ADP-ribosylation domain, indicating involvement in post-translational modifications. Their association implies a functional linkage between transport mechanisms and protein regulation via ADP-ribosylation, potentially in processes like DNA repair or cellular stress responses (49). Cluster_58181 and Cluster_21764 contain the FliK and FliL domains, respectively. Although no direct evidence links them functionally to the T6SS, studies have shown that T6SS can modulate flagellar gene expression. For example, in *Citrobacter freundii* strain CF74, deletion of T6SS genes resulted in decreased expression and secretion of the flagellin protein FliC (50).

Several known T6SS structural components, such as *tssK* and *tssL*, are not significantly associated with T6SS-Eb1. Some components are not significantly associated with T6SS, suggesting that they play roles distant from T6SS. The four most prevalent orphan clusters identified are associated with flagellar proteins, including Cluster_72628 (FlhE), Cluster_42706 (FliH), Cluster_75477 (FliO), and Cluster_87477 (FliT), which appear 1,004, 912, 910, and 907 times, respectively. Notably, in *Edwardsiella tarda*, deletion of the flagellar protein FlhC significantly reduces the secretion of the T6SS effector protein EvpC when *fliC* is deleted (51). This suggests a yet unexplored link between flagellar proteins and the T6SS in *Enterobacter*.

In addition to those genes with known functions, we also identified seven genes with unknown functions that are likely to represent putative T6SS effectors, and five of them have a cognate immunity protein encoded adjacent (Fig. 5D). Particularly notable are pair clusters involved in toxin-antitoxin activity, such as Locus3, which consists of Cluster_70459 linking Cluster_107788, which is likely a potential E-I pair that has not yet been discovered. Locus 6 consists of Cluster_9222 and Cluster_47844, with Cluster_9222 annotated as a T6SS effector by Bastion6. This locus contains two TonB domains and a Plug domain. The TonB domains are associated with the uptake and transport of large substrates, such as iron-siderophore complexes and vitamin B12, while the Plug domain functions as an independently folding subunit of TonB-dependent receptors, opening the channel upon ligand binding. These findings suggest that T6SS-Eb1 may also play a role in iron acquisition. Cluster_59661 links Cluster_3274, which is stated as T6SE in Locus5. This locus contains a *Caudovirales* tail fiber assembly protein domain (PF02413), DUF3751 (PF12571), Phage_fiber_2 (PF03406), and Gp34_C (PF21446), commonly associated with bacteriophage tail fibers, suggesting roles in host recognition and attachment. These proteins may mimic phage tail fibers to facilitate interactions with bacterial cells, potentially aiding in the delivery of toxic effectors or in evading bacterial defenses (52). Clusters annotated as T6SEs that contain domains of unknown function (DUFs) are also noteworthy. For example, Locus1, Locus2, and Locus7 may function as independent effectors or act together with their linked clusters as E-I pairs within the T6SS. Overall, this study expands our understanding of T6SS function in *Enterobacter*. While T6SS-Eb1 is likely involved in immobilizing competitive bacteria, it may also participate in non-canonical functions, such as metal ion acquisition and adhesion. However, we note that many of these functional inferences are based on domain annotations or computational predictions. Experimental validation will be

required to confirm the roles of these putative T6SS effectors. All the TAOG annotations are in the supplementary (Table S4).

## DISCUSSION

T6SS is a nanomachine utilized by bacteria for bacterial competition, host interaction, and environmental adaptation, with extensive studies conducted in species such as *Pseudomonas*, *Vibrio*, and *Burkholderia* (8). However, T6SS is still understudied in *Enterobacter*, a genus known for its clinical relevance and ecological diversity. To fill this gap, we performed a comprehensive genomic survey, uncovering the diversity, distribution, and functional associations of T6SSs across over 4,800 *Enterobacter* genomes. In *Enterobacter*, multiple T6SSs are present, and strains lacking these systems are extremely rare, making it difficult to identify novel effectors solely through comparative genomics. Previous studies neither provided a genus-wide survey of T6SSs nor systematically examined the associated effectors (53). By comprehensively identifying T6SSs across all species within the genus and assessing correlations between homologous protein families and T6SSs using P/A values, we identified three distinct T6SSs and seven effectors or E-I pairs.

Based on gene content and synteny, we classified three main types of T6SS loci of *Enterobacter* spp., which are designated as T6SS-Eb1, T6SS-Eb2, and T6SS-Eb3 in this study (Fig. 1A). These T6SS systems differ not only in having distinct gene organization but have distinct distributions among species, indicating divergent evolutionary pathways: T6SS-Eb2 is universally distributed in both basal and derived *Enterobacter* lineages, such as environmental species, such as *E. lignolyticus*, *E. soli*, and *E. oligotrophicus* (25, 54, 55), in which T6SS-Eb1 is largely undetectable. The distribution pattern suggests that T6SS-Eb2 is an evolutionarily conserved and possibly ancestral system in the genus. Its conservation in basal lineages and environmental isolates further implies that it may provide functions essential for survival in soil, plant-associated, or lignocellulose-rich environments. T6SS-Eb1, however, is more frequently observed in clinically associated species, such as *E. cloacae* and *E. hormaechei*, indicating it may represent a more recently acquired or expanded locus associated with host adaptation or pathogenicity. Previous studies have explored the function of T6SS-Eb2, particularly in *E. cloacae* strain ATCC 13047 (30), where two T6SSs (we name them T6SS-Eb1 and T6SS-Eb2 in this study) have shown involvement in bacterial competition and host colonization, respectively. In this strain, a *clpV1* variant (*tssH* from T6SS-Eb1 in this study) impaired interbacterial killing, while *clpV2* (*tssH* from T6SS-Eb2 in this study) was involved in biofilm formation and adherence. However, our genomic analysis indicates that the T6SS-Eb2 locus in ATCC 13047 lacks the core component TssD, whereas most T6SS-Eb2 loci across the genus harbor all 13 core components. This disparity suggests that ATCC 13047 may be a partially functional variant and that T6SS-Eb2 may have a more general and potentially specific ecological role than is indicated in this specific strain. Together with prior findings that T6SS-Eb1 is more prevalent in pathogenic lineages and linked to bacterial antagonism (18), these findings add to evidence supporting a functional specialization model where T6SS-Eb1 is optimized for interbacterial competition within host-related environments, and T6SS-Eb2 is specialized to play more versatile environmental adaptation functions.

Multiple *vgrG*s located in the main T6SS cluster or *vgrG* islands can be employed by one T6SS as previously described (56). We observed an increasing number of *vgrG* genes in genomes containing complete T6SS loci compared to genomes without any T6SS loci (Fig. 2A), indicating their expansion alongside T6SS acquisition. However, in strains that already possess multiple T6SS loci, this increase is less pronounced, potentially due to functional redundancy. Only 7,412 proteins downstream of the 14,812 identified *vgrG*s have been detected as effectors, indicating that the downstream region of *vgrG* remains a largely unexplored reservoir of T6SS effectors. On the other hand, many identified effectors are not located downstream of *vgrG* (108306/7412), suggesting the existence of additional unexplored effector reservoirs.

T6SS effectors are typically accompanied by immunity proteins to protect the host from self-intoxication (57). In *Enterobacter*, we identified 76 effector families and 36 immunity protein families, indicating a non-equivalent distribution. The diversity of T6SS effectors is an important reason for the abundance of effectors (33). But this asymmetry may result from cross-protective immunity or non-immunity-mediated protection mechanisms. For example, the T6SS effector TseH identified in *Vibrio cholerae* can lyse *Aeromonas* and *Edwardsiella* species but is non-toxic to *E. coli*. The reason is that envelope stress responses protect *E. coli* against T6SS-delivered TseH (58). Recent studies have also demonstrated that other non-immunity-mediated protection mechanisms, such as exopolysaccharides produced by *V. cholerae*, can protect against exogenous T6SS attacks from different bacterial species (59). Additionally, non-specific immunity proteins and effectors targeting non-bacterial cells can contribute to this phenomenon. For instance, in *S. marcescens*, two anti-fungal T6SS effector proteins, Tfe1 and Tfe2, have been identified, while no anti-bacterial toxicity has been detected for Tfe1 and Tfe2 upon overexpression, nor has any candidate immunity gene been identified (60). Recent studies investigating the mechanisms of Tfe1 and Tfe2 in *S. marcescens* also indicate that these two effectors lack corresponding immunity proteins (61).

Previous studies have primarily identified effectors by analyzing proteins downstream of *vgrG* or PAAR within T6SS loci. However, this approach overlooks auxiliary modules containing secreted T6SS components and their downstream-encoded effectors, as well as orphan effectors that are not located downstream of *vgrG*. Our method, previously validated in *V. parahaemolyticus*, successfully predicted T6SS effectors by identifying DUF4225-encoding genes and their adjacent downstream genes as antibacterial T6SS E-I pairs (62). This approach is not only high-throughput but also identifies not just effector and immunity proteins but also previously unrecognized T6SS-associated proteins. To identify genes consistently associated with T6SS, we defined T6SS-associated orthologous groups as TAOGs, based on statistical enrichment in T6SS+ vs T6SS− genomes.

This approach shows the known effectors, regulators, and immunity proteins in strains with a specific complete T6SS significantly higher than those without this type of complete T6SS (Fig. 4), validating the robustness of this methodology. An exception was seen in *E. hormaechei*, where strains with complete T6SS-Eb2 showed slightly fewer immunity proteins (Fig. S7C). This may be due to the confounding presence of T6SS-Eb1, a dominant system in the genus, or reflect the conservation of immunity genes even in strains lacking complete offensive modules. Notably, TssK abundance is not significantly different between T6SS+ and T6SS− genomes, suggesting its role in serving additional cellular functions beyond T6SS.

Some TAOGs were enriched in flagellar-related genes, such as *fliC*, *fliK*, and *flhE*, which were more prevalent in T6SS+ genomes, and some of them were annotated as T6SE. One possibility is that both systems contribute to interbacterial competition, and bacteria with stronger competitive abilities tend to retain both. For instance, in *Citrobacter freundii*, deletion of T6SS components decreased the production and secretion of FliC, resulting in reduced motility. This indicates that a functional T6SS is necessary for the proper regulation of flagellar assembly and function (50). The absence of key T6SS components has been shown to impact the structural integrity of flagella. In *Pseudomonas fluorescens*, mutants lacking T6SS structural proteins, such as *hcp1* or *tssC*, were observed to be non-motile and lacked flagellar filaments, suggesting that the assembly or maintenance of flagella depends on a complete T6SS apparatus (63). Flagellar proteins annotated as T6SEs—such as FliK (Cluster_21764), which is genetically linked to FliL (Cluster_58181; Fig. 5C)—may function as secreted proteins and therefore share characteristics with T6SS effectors, leading to their classification as T6SEs by the machine-learning model. Despite direct evidence for FliK serving as a secreted effector being lacking, its evolutionary association with T6SS components makes it an attractive candidate for further study.

In summary, we provide a systematic characterization of T6SS in *Enterobacter*, revealing three major types with distinct genomic features and distributions. Our

definition of T6SS-associated orthologous groups enables the discovery of novel effectors, immunity proteins, and accessory loci. This work lays a foundation for future studies into how T6SS contributes not only to microbial competition but also to environmental adaptation, stress responses, and potential host interactions in *Enterobacter* spp.

## MATERIALS AND METHODS

### Data acquisition and quality assessment

In this analysis, 9,490 completely sequenced *Enterobacter* genomes were selected and downloaded from the NCBI FTP (ftp.ncbi.nlm.nih.gov/genomes/genbank/bacteria/) site in October 2023. To avoid bias generated by data contamination and incompleteness, we performed data cleaning. A total of 4,805 genomes were retained, with completeness greater than 99% and contamination less than 1%, as evaluated by CheckM v1.2.2 (64). All high-quality *Enterobacter* genomes remaining were annotated as protein sequences using Prokka v1.13 (65) with default settings. All genomes were classified into species in accordance with the NCBI download directory. Although some studies suggest that *Enterobacter aerogenes* should be classified under the genus *Klebsiella* rather than *Enterobacter* (66), we have included it in our analysis to ensure comprehensiveness. GTDB-Tk v2.1.0 (67) was used to determine the taxonomy of all filtered genomes, and the results were consistent with the species-level classifications from NCBI.

### Identification and classification of T6SS components, effectors, and immunity proteins in *Enterobacter*

All known T6SS component protein sequences, as well as T6SS-associated protein sequences (including effectors, regulators, and immunity proteins), were downloaded from the SecReT6 Database (https://bioinfo-mml.sjtu.edu.cn/SecReT6/) in December 2023. Proteinortho v6.2.3 (68) was used to analyze 4,805 *Enterobacter* genomes alongside T6SS-related proteins to identify homologous proteins of T6SS core components in *Enterobacter*. Two core components were considered adjacent if fewer than five intervening genes were present between them, except for TssI, which is more closely associated with variable regions containing T6SS effectors, for which a more lenient threshold (<10 intervening genes) was applied.

A locus was classified as complete if it contained all 13 core components, with each component neighboring at least one other main component. A locus was deemed partial if it lacked at least one but no more than ten main components, provided the remaining core components were adjacent. Loci with fewer than three neighboring core components were not classified as T6SS loci.

Loci with core components arranged in the same order were classified as the same type of T6SS, which means two loci were considered different if the arrangement of the 13 structural components (TssA-TssM) differed. Additionally, if a locus contained most of the 13 structural components but either lacked one or included an additional main component compared to the dominant T6SS loci, it was classified as a subtype of the dominant T6SS, and we will only include it in the analysis if the number of genomes containing this locus exceeds 1% of the total number of genomes.

### Phylogenetic construction

The TssB phylogenetic tree was constructed using at least one TssB protein sequence from each T6SS subtype downloaded from the SecReT6 Database (https://bioinfo-mml.sjtu.edu.cn/SecReT6/), along with TssB sequences from the three T6SS types in *Enterobacter* (T6SS-Eb1, T6SS-Eb2, and T6SS-Eb3), totaling 27 sequences. The sequences were first aligned using MAFFT (69), then trimmed to remove gaps using TrimAl (70), and finally used as input for IQ-TREE v2.2.2.3 (71), where a fast likelihood tree search was performed using the LG+I+G model. For the T6SS-Eb3 phylogeny, all experimentally validated TssB sequences from *Enterobacteriaceae* available in the SecReT6 Database

were collected, and the tree was reconstructed following the same procedure described above, also employing the LG+I+G model. For the T6SS-Eb3 multi-locus phylogenetic analysis, experimentally validated T6SS loci from *Enterobacteriaceae* were retrieved from the SecReT6 database. Loci containing more than 10 core components were retained, and five shared core components (TssA, TssG, TssJ, TssK, and TssL) were identified. Each protein was aligned using MAFFT, trimmed with trimAl, concatenated into a supermatrix, and analyzed using IQ-TREE under the LG+I+G model.

One strain was randomly selected from each of the 23 species within the *Enterobacter* genus after quality control, with *E. coli* (ATCC:11775) designated as the outgroup. Protein sequences were used as the input format for PhyloPhlAn v3.0.67 (72) to identify marker genes and perform multiple sequence alignment. IQ-TREE v2.2.2.3 (71) was utilized on the aligned sequence and performed a fast likelihood tree search using the LG+I+G model. Another TssB phylogenetic tree was also constructed via iqtree and using the Q. yeast+G4 model. Two tree files were uploaded to ITOL (https://itol.embl.de) for visualization.

## Identification and characterization of VgrG proteins in *Enterobacter*

We collected all VgrG protein sequences and used Pfam_scan v1.6 (73) to identify their corresponding HMM profiles. Subsequently, we constructed an HMM database using HMMER3 (74) to detect all VgrG proteins across individual *Enterobacter* strains.

## Statistical analysis

We applied pairwise correlation analysis using $2 \times 2$ contingency tables to test whether the three major T6SS loci are statistically independent. In this analysis, we used both the chi-square test statistic and the Phi ($\varphi$) coefficient to assess the strength and significance of the association between these loci. Because T6SS-Eb1 and T6SS-Eb2 are the primary T6SS loci in *Enterobacter*, and the number of T6SS-Eb3 instances was too small to yield statistical power, we divided all genomes into four groups: those carrying both loci, those carrying only one of the loci, and those lacking a complete T6SS. We then used the nonparametric Kruskal-Wallis test to assess whether there were significant differences among these four groups in the number of *vgrG* genes, implemented in R via the kruskal.test() function.

A nonpaired Student's *t*-test (stats R package) was applied for the significance test of the number of effectors, regulators, and immunity proteins per genome. For T6SS-Eb1 and T6SS-Eb2, let P be the number of strains with complete T6SS-Eb1 or T6SS-Eb2, and A be the number of strains with no T6SS-Eb1 or T6SS-Eb2. We don't concern ourselves with the strains with partial T6SS, which contain more than three components but fewer than 13 core components. All proteins belonging to 4,805 *Enterobacter* spp. genomes are divided as groups via CD-HIT v4.6 (75) with 80% coverage and 80% similarity (76). After this step, we clustered all of the proteins by sequence identity, and we can preliminarily infer that proteins within the group have similar functions based on their sequence similarity. For each group, out of consideration for strains with partial T6SS, let *Pc* be the number of proteins present in strains with complete T6SS, and let *Ac* be the number of strains with no T6SS. The +1 in the denominator prevents division by zero and reduces bias from small sample sizes, ensuring a stable and reliable calculation of the P/A value. The P/A value is calculated as

$$\text{P/A value} \ = \frac{Pc \cdot A}{(Ac + 1) \cdot (P + 1)} \, .$$

Protein clusters associated with the two types of T6SS were calculated separately, and the significance of the calculated ratios was evaluated using the chi-square test. Proteins linked to each other transfer to clusters linked to each other and are visualized via the Cytoscape program (v3.9) (77).

## Identification of putative T6SS effectors and their immunity proteins in TAOG

We identified putative T6SS effectors by uploading protein sequences and downloading the prediction results in a text file from Bastion6, a two-layer SVM-based ensemble model used to identify potential T6SEs (78).

## ACKNOWLEDGMENTS

The authors express their gratitude to the Bioinformatics Platform at The First Hospital of Jilin University for its invaluable support and provision of computational resources.

The project was supported by the Jilin Provincial Science and Technology Development Project of Prof. Qingtian Guan, grant number YDZJ202401276ZYTS.

Z.P.: Formal analysis; Validation; Investigation; Visualization; Methodology; Writing—original draft. N.Z.: Formal analysis; Visualization; Writing—review and editing. W.Y.: Formal analysis. L.J.: Formal analysis. T.D.: Formal analysis. R.W.: Software. S.J.: Software. X.G.: Software. Z.L.: Conceptualization; Supervision; Writing—review and editing. Q.G.: Conceptualization; Supervision; Methodology; Writing—original draft; Writing—review and editing.

## AUTHOR AFFILIATIONS

[1]Bioinformatics Laboratory, Center for Infectious Diseases and Pathogen Biology, The First Hospital of Jilin University, Changchun, China

[2]Health Examination Center, The First Hospital of Jilin University, Changchun, China

[3]The Core Facility of the First Hospital of Jilin University, Changchun, China

[4]Department of Respiratory Medicine, Center of Infectious Diseases and Pathogen Biology, Key Laboratory of Organ Regeneration and Transplantation of the Ministry of Education, State Key Laboratory for Diagnosis and Treatment of Severe Zoonotic Infectious Diseases, Key Laboratory for Zoonosis Research of the Ministry of Education, Jilin Provincial Key Laboratory for Individualized Diagnosis and Treatment of Pulmonary Diseases, The First Hospital of Jilin University, Changchun, China

## AUTHOR ORCIDs

Zhihan Peng http://orcid.org/0009-0003-2259-3484
Zhaoqing Luo http://orcid.org/0000-0001-8890-6621
Qingtian Guan http://orcid.org/0000-0001-5903-1774

## FUNDING

| Funder | Grant(s) | Author(s) |
| --- | --- | --- |
| 吉林省科学技术厅 \| Jilin Provincial Scientific and Technological Development Program (Jilin Scientific and Technological Development Program) | YDZJ202401276ZYTS | Qingtian Guan |

## AUTHOR CONTRIBUTIONS

Zhihan Peng, Conceptualization, Formal analysis, Methodology, Visualization, Writing – original draft | Ning Zhu, Formal analysis, Visualization, Writing – review and editing | Wenjing Yi, Formal analysis | Lili Jiang, Formal analysis | Tingting Dong, Formal analysis | Ruihong Wu, Software | Shanshan Jia, Software | Xiaohan Guo, Software | Zhaoqing Luo, Conceptualization, Supervision, Writing – review and editing | Qingtian Guan, Conceptualization, Methodology, Supervision, Writing – original draft, Writing – review and editing

## DATA AVAILABILITY

The genome assembly summary and the phylogenetic tree files used in this study are available in Zenodo with the DOI 10.5281/zenodo.15305012. The source code is available at https://github.com/Guan-biolab/Enterobacter-TAOG-Finder.

## ADDITIONAL FILES

The following material is available online.

### Supplemental Material

**Supplemental material (mSystems01781-25-s0001.docx).** Figures S1 to S7; captions for Tables S1 to S4.
**Table S1 (mSystems01781-25-s0002.xlsx).** Taxonomic assignment of genomes based on GTDB-Tk analysis.
**Table S2 (mSystems01781-25-s0003.xlsx).** Pairwise associations of T6SS loci in *Enterobacter*.
**Table S3 (mSystems01781-25-s0004.xlsx).** BLAST alignment of T6SS-Eb3-containing contigs against the NCBI non-redundant database.
**Table S4 (mSystems01781-25-s0005.xlsx).** Annotation of T6SS-Eb1-associated and T6SS-Eb2-associated orthologous groups.

### Open Peer Review

**PEER REVIEW HISTORY (review-history.pdf).** An accounting of the reviewer comments and feedback.

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
