## [Reviewer comments · mSystems]

Pan-Genome Insights into Type VI Secretion Systems and Their Functional Repertoires in *Enterobacter*

Zhihan Peng, Ning Zhu, Wenjing Yi, Lili Jiang, Tingting dong, Ruihong Wu, Shanshan Jia, Xiaohan Guo, Zhaoqing Luo, and Qingtian Guan

Corresponding Author(s): Qingtian Guan, The First Hospital of Jilin University

Review Timeline:

Submission Date:

December 19, 2025

Accepted:

March 4, 2026

Editor: Andrew Bartko

Reviewer(s): Disclosure of reviewer identity is with reference to reviewer comments included in decision letter(s). The following individuals involved in review of your submission have agreed to reveal their identity: Min Yue (Reviewer #2)

Transaction Report:

DOI: <https://doi.org/10.1128/msystems.01781-25>

Re: mSystems01781-25 (**Pan-Genome Insights into Type VI Secretion Systems and Their Functional Repertoires in *Enterobacter***)

Dear Dr. Qingtian Guan:

Your manuscript has been accepted, and I am forwarding it to the ASM production staff for publication. Your paper will first be checked to make sure all elements meet the technical requirements. ASM staff will contact you if anything needs to be revised before copyediting and production can begin. Otherwise, you will be notified when your proofs are ready to be viewed.

Sincerely,
Andrew Bartko
Editor
mSystems

Reviewer #2 (Comments for the Author):

None